# One-Carbon Metabolism in Alzheimer’s Disease and Parkinson’s Disease Brain Tissue

**DOI:** 10.3390/nu14030599

**Published:** 2022-01-29

**Authors:** Karel Kalecký, Paula Ashcraft, Teodoro Bottiglieri

**Affiliations:** 1Institute of Biomedical Studies, Baylor University, Waco, TX 76712, USA; Karel_Kalecky@baylor.edu; 2Center of Metabolomics, Institute of Metabolic Disease, Baylor Scott & White Research Institute, Dallas, TX 75204, USA; paula.ashcraft@bswhealth.org

**Keywords:** one-carbon metabolism, homocysteine, betaine, Alzheimer’s disease, Parkinson’s disease, dementia, levodopa, metabolomics, brain frontal cortex

## Abstract

Disruptions in one-carbon metabolism and elevated homocysteine have been previously implicated in the development of dementia associated with Alzheimer’s disease (AD) and Parkinson’s disease (PD). Moreover, a PD diagnosis itself carries substantial risk for the development of dementia. This is the first study that explores alterations in one-carbon metabolism in AD and PD directly in the human brain frontal cortex, the primary center of cognition. Applying targeted liquid chromatography–tandem mass spectrometry (LC-MS/MS), we analyzed post-mortem samples obtained from 136 subjects (35 AD, 65 PD, 36 controls). We found changes in one-carbon metabolites that indicate inefficient activation of cystathionine β-synthase (CBS) in AD and PD subjects with dementia, the latter seemingly accompanied by a restricted re-methylation flow. Levodopa–carbidopa is known to reduce available vitamin B6, which would explain the hindered CBS activity. We present evidence of temporary non-protein-bound homocysteine accumulation upon levodopa intake in the brain of PD subjects with dementia but not in non-demented PD subjects. Importantly, this homocysteine elevation is not related to levodopa dosage, disease progression, or histopathological markers but exclusively to the dementia status. We hypothesize that this levodopa-induced effect is a direct cause of dementia in PD in susceptible subjects with reduced re-methylation capacity. Furthermore, we show that betaine best correlates with cognitive score even among PD subjects alone and discuss nutritional recommendations to improve one-carbon metabolism function.

## 1. Introduction

One-carbon metabolism occupies a unique place at the intersection of epigenetic and metabolic regulation and nutrition (intake of proteins and B vitamins). Homocysteine (Hcy) is a central metabolite formed as an intermediate product of the one-carbon metabolism following trans-methylation (see Figure 1 for overview) and is either metabolized into cystathionine and cysteine (Cys; B6-dependent) or recycled back into methionine via methionine synthase (MTR; B2,6,9,12-dependent) or betaine-homocysteine methyltransferase (BHMT; active mostly in liver and kidneys but to a certain degree also in the brain). Hcy accumulation, due to genetic factors or nutritional inadequacy, leads to pathological consequences. Mechanistically, Hcy alters protein structure by the homocysteinylation of lysine residues via homocysteine thiolactone [1] and also activates NMDA (N-methyl-D-aspartate) receptors [2]. In severe cases of inborn errors of Hcy metabolism, the symptoms range from endothelial dysfunction through connective tissue impairment and up to mental retardation [3].

Elevation of plasma total Hcy is associated with a higher risk of atherosclerosis and vascular disease, especially ischemic stroke [4,5,6]. Increased plasma total Hcy is also an independent risk factor for dementia and Alzheimer’s disease (AD) [7] and shows a mild correlation with the mini-mental state examination (MMSE) cognitive score [8] among subjects with AD [9].

The diagnosis of Parkinson’s disease (PD) is a substantial risk factor for developing dementia [10]. The onset of PD may not be directly related to Hcy or B vitamins [11,12,13]; however, the anti-parkinsonian drug levodopa, which is converted into dopamine in the brain, undergoes partial deactivation by methylation via catechol-O-methyltransferase (COMT) and increases plasma total Hcy levels [14]. Increased levodopa dosage has also been reported as a risk factor for developing dementia [15]. Recently, a randomized controlled trial showed an association between elevated plasma total Hcy and faster decline in MMSE score among subjects with PD [16].

Despite this evidence implicating Hcy in AD and PD dementia, so far, all investigations have focused mainly on human plasma or animal models. No studies have explored one-carbon metabolism in these disorders directly in the human brain tissue. The closest such investigation performed involved several Hcy-related measures and univariable analysis found reduced homocysteine-thiolactonase activity in AD cortex but no difference in total Hcy, although the cohorts were unbalanced in terms of sex [17]. In another study, human brain cortex tissue from controls and patients with AD were analyzed for S-adenosylmethionine (SAM) and S-adenosylhomocysteine (SAH) only. Interestingly, levels of SAH correlated with phenylethanolamine N-methyltransferase and COMT activities and other markers of neurodegeneration and cognitive function [18].

Interrogating brain tissue, especially in frontal cortex—the primary center of cognition and its pathologies, can help us understand the involvement of one-carbon metabolism in cognitive impairment in AD and PD and direct us toward possible beneficial strategies for the reduction or prevention of cognitive decline.

In this study, we performed targeted mass spectrometry analysis of one-carbon metabolism in brain frontal cortex samples of subjects with AD dementia, subjects with PD across spectrum of cognitive impairment, and cognitively normal controls. The results suggest a reduced activity of cystathionine β-synthase activity (CBS) in AD and PD with dementia, in the latter case, seemingly accompanied by reduced re-methylation flow via MTR not attributable to folate status. We discovered that Hcy is elevated in the brain tissue of PD subjects with dementia upon levodopa delivery but not in non-demented PD subjects. Additionally, we found that betaine is associated with MMSE even among PD subjects alone.

## 2. Materials and Methods

### 2.1. Study Subjects

Post-mortem brain frontal cortex samples (500 mg) of 35 AD cases, 65 PD cases at various stages of cognitive impairment (32 with dementia and 33 without dementia subdivided into 19 with mild cognitive impairment (MCI) and 14 cognitively normal), and 36 cognitively normal controls were obtained from the Banner Sun Health Research Institute brain bank (Sun City, AZ, USA) [19]. The samples were collected between 2004 and 2018 from deceased donors and continuously stored at −80 °C. The diagnosis followed histopathological examination and NIA–Reagan classification [20], for which the AD cases met “high likelihood of AD”. PD subjects had two of the three cardinal clinical signs of resting tremor, muscular rigidity, and bradykinesia, along with pigmented neuron loss and Lewy bodies’ presence in substantia nigra, and were treated with levodopa–carbidopa medication. The status of dementia and MCI correspond to clinical diagnosis. Controls were without a history of cognitive impairment and parkinsonism. All subjects were White Americans with no other major pathology of central nervous system.

All sample handling was done on dry ice to avoid multiple freeze-thaw cycles. Post-mortem intervals as well as duration of storage in freezer at −80 °C were controlled for in the analysis.

### 2.2. Chromatography and Mass Spectrometry

We performed targeted liquid chromatography–tandem mass spectrometry analysis integrating three separate assays. Folate intermediates were analyzed after deconjugation to monoglutamate forms following a recently published protocol [21]. Non-protein-bound Hcy and Cys as well as dihydroxyphenylalanine (DOPA) were measured using the Biocrates MxP Quant 500 kit (Biocrates Life Sciences AG, Innsbruck, Austria) after extraction in ethanol with phosphate-buffered saline (85:15). Metabolites related to Hcy metabolism were measured using a method previously described for SAM and SAH [22], extended with additional metabolites (Q1/Q3 mass (*m/z*), declustering potential (V) entrance potential (V), collision energy (V), collision cell exit potential (V): methionine 150.1/104.1, 40, 10, 14, 10; cystathionine 223.1/134.1, 40, 10, 18, 12; choline 104.2/45.2, 75, 10, 36, 2; betaine (trimethylglycine) 118.1/59.2, 77, 10, 27, 2; dimethylglycine 104.1/58.1, 40, 10, 17, 8) and their respective labeled standards. Tissue was prepared by sonication in 0.4 M perchloric acid at the concentration of 4 µL/mg and further diluted 1:10 with internal standards in purified water.

The analysis was run on a Shimadzu Nexera chromatography platform (Shimadzu Corporation, Kyoto, Japan) coupled to a Sciex QTrap 5500 mass spectrometer (AB Sciex LLC, Framingham, MA, USA), except for the folate assay, which was run on a Waters Xevo TQ MS spectrometer (Waters Corporation, Milford, MA, USA).

Samples were evenly randomized prior to processing to avoid any accidental bias toward one of the groups of interest. Each batch included at least seven-point calibration and repeats of a quality control sample (typically achieving a coefficient of variation <10%). Chromatographic peaks were manually reviewed, and areas of metabolites were coupled to areas of their respective internal standards.

### 2.3. Data Preprocessing

Several metabolites were not readily detected in the brain tissue and were excluded from the analysis, specifically dimethylglycine and all folate intermediates except for tetrahydrofolate (THF) and 5-methyltetrahydrofolate (MTHF).

To better approximate Gaussian distributions, we applied Box–Cox transformation on the concentration values with R package *car* [23]. Outliers were detected and adjusted with conventional Tukey’s fencing [24] to protect against skewing the means by extreme values while not reducing the variance greatly compared to outlier removal. Finally, the values were standardized with respect to control samples to facilitate comparison of regression coefficients in the statistical analysis.

Since the regression requires all covariates to be non-missing, four missing body mass index (BMI) values were imputed as a mean value conditional on the diagnosis group and sex. Two subjects of unknown apolipoprotein E (*APOE*) genotype were regarded as non-ε4 carriers. Among measured values, we found no THF peak in four samples and interpolated these values as half of the least detected THF concentration to avoid strict zeros.

### 2.4. Statistical Analysis

Key characteristics of subjects in each diagnostic group were compared with Fisher’s exact test for binomial variables and analysis of variance (ANOVA) on a linear model constructed with the R package *nlme* [25] for continuous variables, not assuming equivalence of variance among the diagnosis groups.

The differential analysis was based on a series of multivariable linear regression models with R package *nlme* [25], one for each metabolite, with the measured values as dependent variables and diagnosis groups as independent variables, without the assumption of equivalence of variance among the diagnosis groups. The models included covariates for age, sex, BMI, count of *APOE* ε4 alleles, multivitamin supplementation and supplementation of vitamin B12 (both of which can have a direct impact on the one-carbon metabolism), post-mortem collection interval, and the total length of storage in freezer. The last two covariates were log-transformed to be able to capture any time-related exponential decay. Due to standardization, the regression coefficients have a unit of 1 standard deviation of the distribution of controls.

Assessment of collinearity among all regressors was based on the magnitude of Pearson’s correlation coefficients and adjusted generalized variable inflation factor (GVIF) calculated with the R package *car* [23]. We found no evidence of significant collinearity (all Pearson’s |r| < 0.4 and adjusted GVIFs < 1.5).

For each diagnostic group, two-tailed *p*-values across the models were obtained and controlled for false discovery rate (FDR) using the Benjamini–Hochberg procedure [26]. Effects with FDR ≤ 0.05 are considered statistically significant.

We performed interaction analysis for levodopa medication in PD. The medication status of study subjects at the time of their departure is unknown, and therefore, we measured DOPA concentrations in the samples. From these measurements, we constructed histograms for control subjects and PD patients. We identified a heuristic threshold for DOPA values corresponding to active medication based on a notch in the histogram for PD samples, which corresponded to the more than 95% quantile in controls. The interactions of levodopa with individual PD groups were then modeled as independent variables.

Association analysis for MMSE followed a procedure identical to that of the differential analysis, with the independent variable being MMSE instead of diagnosis groups, and only subjects with known MMSE were considered. This means that all covariates were still included. We repeated the analysis with PD samples only and, further, with adjustment for interactions with levodopa as above.

Correlation analysis was performed on all pairs of metabolites with Hcy as the main compound of interest due to its adverse effects and, additionally, with all pairwise combinations between betaine, methionine, THF, and MTHF to capture a potential cross-talk between Hcy re-methylation pathways. We included the SAM/SAH calculated ratio for its pathophysiological utility [27], similarly to the case for THF/MTHF. Pearson’s correlation coefficients were computed for each diagnostic group and compared with the control group using Fisher’s z-transformation. In this analysis, PD subjects on levodopa medication at the time of death as identified in the interaction analysis were excluded to reduce medication-caused disturbances in the metabolic fluxes unless stated otherwise.

## 3. Results

### 3.1. Subject Characteristics

Important subject and sample-related characteristics are summarized in Table 1. Sex and age were reasonably balanced between the groups. Note the substantially higher prevalence of *APOE* ε4 carriers among AD subjects. Individual PD groups were reflected by progressively worsening MMSE score. The diagnosis duration is significantly higher among PD subjects with dementia, which also corresponds to the disease progression in the motor section of unified Parkinson’s disease rating scale (UPDRS-M), whereas PD subjects with MCI and cognitively normal PD subjects had very similar characteristics. Histopathological density scores of neurofibrillary tangles and senile plaque, as evaluated using templates of the Consortium to Establish a Registry for Alzheimer’s Disease (CERAD) [28], are markedly higher in AD by definition. PD subjects exhibit a mildly higher tangle score than controls, with no difference among individual PD subgroups. The plaque score is mildly elevated only in non-demented PD subjects but not in PD subjects with dementia. The presence of Lewy bodies measured by Unified Lewy body stage [29] is characteristic of PD and is higher in PD subjects with MCI and dementia than in cognitively normal PD subjects.

Cognitively normal PD subjects reported multivitamin supplementation twice as frequently, and this effect is controlled for in the analysis. Post-mortem intervals between death and tissue storage are very short, less than 3.5 h on average, and identical across all groups. Duration of storage in the freezer since the collection is somewhat higher for the control samples, as we had to go further in time and obtain “older” samples to get an adequate number of controls. Nevertheless, it does not pose an issue: any tissue degradation would be reflected in higher levels of choline and amino acids, but the observed values do not conform to this pattern. In addition, the choline levels of the longer stored control samples were indistinguishable from those of control samples with shorter freezer storage (Welch’s *t*-test *p*-value = 0.72), which is a sufficient proof of no storage-time-related degradation. Besides, we controlled for a freezer storage effect in the regression alongside other covariates.

All subject metadata together with measured concentrations are provided in Appendix A. The concentrations are also summarized by group in Table 2. The large standard deviations reflect non-normal distributions prior to data transformation.

### 3.2. Differential Analysis

Differential analysis with a linear regression model identified elevated Cys (95% confidence interval (CI_95_) = (0.54; 1.73), FDR = 0.003) and decreased betaine (CI_95_ = (−1.37; −0.35), FDR = 0.007) in the cortex of patients with AD and decreased betaine (CI_95_ = (−1.36; −0.28), FDR = 0.035) in PD subjects with dementia. There are multiple changes with statistically significant *p*-value but less significant FDR: decreased betaine in PD subjects with MCI, increased Cys and SAH in PD subjects with dementia, and increased SAM and decreased choline and THF in AD subjects. The results are listed in Table 3 and further visualized as a group heatmap in Figure 2a.

There is a clear similarity in the profiles of AD group and PD with dementia as well as MCI, often in a progressive manner, which is apparent for Hcy and Cys, betaine and choline, and THF. The disturbance in the methionine cycle for these PD groups seems to stem from the SAH elevation, since the upstream part shows no trend of change, unlike the AD group, where the whole methionine cycle tends to be upregulated. On the contrary, the profile of cognitively normal PD subjects shows few changes and is similar to controls.

### 3.3. Levodopa Interaction

In the next step, we inspected an interaction with PD medication levodopa, for which we heuristically identified PD subjects with abnormally high levels of DOPA compared to controls as explained in the Methods and depicted in Appendix A. This DOPA elevation reflects levodopa medication active in the system at the time of death and we refer to these cases as DOPA+ subjects or subjects following levodopa intake, as opposed to DOPA− subjects without traces of levodopa presence. The results of differential analysis with levodopa interaction are visualized in a group heatmap in Figure 2b. We can see large changes in the PD subjects with dementia on levodopa, which reveal upregulation of SAH (CI_95_ = (0.52; 1.61), FDR ≤ 0.001), Hcy (CI_95_ = (0.82; 1.79), FDR < 0.001), cystathionine (CI_95_ = (0.38; 1.61), FDR = 0.005) and Cys (CI_95_ = (0.27; 1.71), FDR = 0.016) and down-regulation of THF (CI_95_ = (−1.66; −0.49), FDR = 0.002) and MTHF (CI_95_ = (−1.49; −0.16), FDR = 0.028).

Interestingly, cognitively normal PD subjects and those with MCI do not exhibit this interaction with levodopa. This striking elevation of Hcy only in PD subjects with dementia after levodopa intake is captured in Figure 3. The difference is independent of levodopa dosage as well as various measures of disease progression and histopathological markers of neurofibrillary tangles, senile plaque, and Lewy body stage as shown in Figure 4.

### 3.4. MMSE Associations

Further, we explored metabolic associations with MMSE score, which was available for 87% subjects. We detected several significant associations; positive with betaine (CI_95_ = (0.01; 0.07/pt (per MMSE point)), FDR = 0.026) and THF (CI_95_ = (0.01; 0.05)/pt, FDR = 0.034), and negative with Cys (CI_95_ = (−0.08; −0.02/pt), FDR = 0.009), SAM (CI_95_ = (−0.05; −0.01/pt), FDR = 0.035) and Hcy (CI_95_ = (−0.06; −0.01/pt), FDR = 0.035). Details are provided in Table 4.

The results overlap with the trends seen in the AD subjects. Given that the AD group has significantly lower MMSE than the other subjects, there is a possibility that these results reflect associations with AD rather than with MMSE in general. To exclude this possibility, we decided to repeat the analysis only within the PD subjects, which still covers a broad spectrum of cognitive impairment. As a result, only one of the previous associations remained valid and that was with betaine (CI_95_ = (0.03; 0.13/pt), FDR = 0.021). The significance did not change even after accounting for interactions with levodopa. The association of MMSE with betaine is depicted in Figure 5. Notice the relatively steep decrease in mean betaine levels at the beginning of cognitive decline.

### 3.5. Correlation Analysis

Concentrations of metabolites are often correlated and dependent on each other along metabolic pathways. We hypothesized that, in the case of pathway disruptions, such correlations might be considerably weaker, non-existent, or even new ones might appear. To explore this possibility, we performed differential correlation analysis. Surprisingly, we found two pairs with strong correlations in AD subjects that are independent in controls: Hcy and Cys (Pearson’s correlation coefficient (r) r_AD_ = 0.74, r_Controls_ = 0.14, *p* = 0.001, FDR = 0.026), and methionine and THF/MTHF ratio (r_AD_ = −0.63, r_Controls_ = −0.04, *p* = 0.005, FDR = 0.045). In DOPA− PD subjects with dementia, the correlation between Hcy and Cys is similarly strong although with less significant FDR (r_PD-dem_ = 0.69, r_Controls_ = 0.14, *p* = 0.021, FDR = 0.25). Additionally, there seems to be a lack of correlation between Hcy and THF (r_PD-dem_ = −0.05, r_Controls_ = −0.59, *p* = 0.039, FDR = 0.25) and MTHF (r_PD-dem_ = 0.16, r_Controls_ = −0.47, *p* = 0.029, FDR = 0.25), both of which are present in other groups. We did not detect similarly strong changes in PD subjects with MCI and cognitively normal PD subjects even when combined together or including DOPA+ subjects.

The correlations are captured in Figure 6. For the correlation between Hcy and Cys, subjects with AD or PD dementia shared the same slope, while PD subjects with MCI were halfway from controls, suggesting a possible trend of progression.

## 4. Discussion

To the best of our knowledge, this is the first study investigating one-carbon metabolism that includes a comprehensive analysis of metabolites in both the folate and methionine cycle in AD and PD in human post-mortem brain tissue. We have found multiple alterations associated with AD and PD with dementia, sharing some similarities and differences. The discovered alterations are summarized in Figure 7 and further discussed in this section.

### 4.1. One-Carbon Metabolism in Alzheimer’s Disease Dementia

The largest change observed in AD subjects is increased Cys. This elevation is driven by Hcy levels as shown by their significant correlation, which is not present among controls. Conversion of Hcy into cystathionine and Cys is initiated by CBS, followed by cystathionine γ-lyase, both reactions catalyzed by vitamin-B6-dependent enzymes. The activity of CBS is reduced in a normal state and increases upon elevation of SAM, which functions as its allosteric activator [30], to rapidly clear excess Hcy. Indeed, the observed increase in SAM in AD (*p* = 0.046, FDR = 0.1) suggests that this should be the case, although the response may not be sufficient to normalize the concentrations. The methionine cycle shows a trend toward upregulation without any apparent bottleneck and no difference in MTHF. Moreover, we see a new significant correlation between the THF/MTHF ratio and methionine, which indicates that the re-methylation of Hcy via MTR is flowing and methionine levels are driven by the available methylation capacity of MTR. Put together, we see that despite expected activation of CBS, Hcy is not adequately cleared by transsulfuration and continues being recycled via re-methylation up to its limits. This is underlined by significantly decreased betaine, which is used for re-methylation of Hcy via BHMT, and apparently is utilized faster than in controls.

Therefore, the results suggest that the observed changes in one-carbon metabolism in AD may be caused by reduced CBS activity. The Cys upregulation and strong significant correlation between Hcy and Cys can result from reduced flow, however counterintuitive it might appear. This is because a restriction and higher saturation will lead to continuous synchronous flow through the pathway instead of immediate Hcy clearance when needed. The reduced CBS activity could occur from several conditions, including vitamin B6 deficiency, CBS mutation, and downregulated CBS expression. These findings are consistent with reports of increased prevalence of low B6 in plasma among AD subjects [31] and specific CBS mutations associated with AD [32]. Furthermore, reduced CBS activity has been observed in the brains of patients with AD indirectly as reflected by decreased levels of hydrogen sulfide [33], a byproduct in the transsulfuration pathway.

Vitamin B6 deficiency can be detected from blood measurement of this vitamin and easily treated by oral supplementation. In the case or presence of less effective CBS alleles or downregulated CBS expression, a more complex strategy is needed, ideally combining supplementation of vitamin B9, B12, and betaine to increase the capacity of the re-methylation pathways on the one hand and low-methionine diet on the other hand. However, reducing total protein intake in AD subjects might not be desirable due to their tendency to lose lean mass [34]. A randomized controlled trial has already shown reduction in brain atrophy in AD when supplementing vitamins B6, B9, and B12 [35]. Betaine supplementation has also shown clinical benefits [36].

### 4.2. One-Carbon Metabolism in Parkinson’s Disease with Dementia

We found significantly a upregulated transsulfuration pathway in DOPA+ PD with dementia. The upregulation is driven by elevated SAH and Hcy, consistent with the mechanism of SAH influx following levodopa methylation by COMT and generation of Hcy, as SAH and Hcy are interconvertible. Surprisingly, this levodopa-induced increase in Hcy is profound in PD with dementia but did not occur in other PD subjects. We have ruled out that this effect is caused by levodopa dosage, measured DOPA levels, and COMT activity when comparing the ratio of measured DOPA levels in tissue and reported levodopa dose. Neither is the effect related to COMT inhibitors, as they were reported by only one MCI subject and one cognitively normal DOPA+ subject (and these subjects ranked at the high end of Hcy levels in their groups, not lower). Moreover, this effect is present regardless of duration of PD diagnosis, age, UPDRS-M motor score of disease progression, MMSE cognitive score or histopathological markers. This implies that non-demented PD subjects are able to efficiently metabolize levodopa-induced Hcy, whereas PD subjects with dementia have an impaired metabolism, causing Hcy to accumulate in the brain. This is consistent with the trend (albeit non-significant) toward lower Hcy in both groups with non-demented DOPA+ PD subjects, as an expected after-effect upon activation of CBS and rapid clearance of Hcy, as opposed to its accumulation in PD subjects with dementia. Investigating DOPA− PD subjects with dementia, we can see a significant correlation among Hcy and Cys as in AD, but not in non-demented DOPA− PD subjects or controls, suggesting a shift toward the transsulfuration pathway and its possible congestion.

At the same time, we see a lack of correlation between Hcy and MTHF or THF unlike in other groups. In other words, the Hcy level in PD subjects with dementia is almost unaffected by the level of folates. This points toward a restriction in the re-methylation pathway via MTR. Reduced flow through one pathway may exert a higher load on the alternative pathways, which explains the shift toward the transsulfuration pathway, as well as greatly reduced levels of betaine as a result of a higher workload on BHMT, depleting its available re-methylation capacity. Given that the levodopa disruption of the methionine cycle begins with SAH, the restriction in re-methylation would then make it difficult to increase SAM and fully activate CBS to clear the Hcy, explaining its accumulation. We expect that this effect gets enhanced by a low-protein diet, reducing methionine for SAM synthesis. Low-protein diets are recommended for PD patients to improve levodopa absorption [37]. Supplementing with oral SAM could circumvent this metabolic restriction.

The seemingly reduced activity of MTR in PD subjects with dementia is not connected to folates, since folate levels are unchanged for DOPA− cases, and could be a result of deficiency in other related B vitamins (B2, B6, B12) or enzymatic inefficiency (besides MTR, other relevant enzymes are methionine synthase reductase, methylenetetrahydrofolate reductase, methylenetetrahydrofolate dehydrogenase, and serine hydroxymethyltransferase).

Note that the significantly lower levels of THF and MTHF and no longer decreased betaine in DOPA+ PD subjects with dementia are clear evidence of the inhibitory effect of elevated SAM on methylenetetrahydrofolate reductase [38] and BHMT [39] (previously low betaine in DOPA− subjects increased back to normal in DOPA+ subjects). Therefore, increased SAM was also expected to allosterically activate CBS to clear Hcy excess, which we did not observe. Therefore, it appears that there are other factors contributing to the limited CBS activity as in AD, i.e., possible vitamin B6 deficiency, reduced CBS expression, or CBS mutation.

There are no reports of altered CBS activity in Parkinson’s disease. Lower vitamin B6 status does not constitute a risk factor for developing PD [13]. However, lower vitamin B6 has been reported among PD patients [40], also showing some dose-dependent relationship with levodopa [41]. Clinical trials show that levodopa-induced Hcy elevation is affected by vitamins B status [42]. Currently, levodopa medication is typically available in combination with carbidopa, to inhibit peripheral DOPA decarboxylase and improve DOPA delivery across the blood–brain barrier. Carbidopa binds vitamin B6 and can cause its systemic depletion in rats [43]. In subjects who receive the intestinal gel infusion form of levodopa–carbidopa, this treatment is frequently connected to peripheral neuropathy [44], although the oral form has been implicated in some cases as well [45]. We hypothesize that the indication of limited CBS activation that we observe in the brains of PD subjects with dementia is caused by a temporal medication-induced vitamin B6 deficiency, and together with the impaired flow through MTR, this leads to a temporal levodopa-induced elevation of Hcy that may contribute to the etiology of dementia.

### 4.3. One-Carbon Metabolism in Parkinson’s Disease without Dementia

Cognitively normal PD subjects and PD subjects with MCI do not exhibit the levodopa-induced elevation of SAH and Hcy in brain tissue. We do not see any evidence of restricted flow in the re-methylation pathways or changes in the transsulfuration pathway, which could be the reason why Hcy does not accumulate as in PD-dem subjects.

In DOPA− cases with MCI, the correlation of Hcy and Cys is not significant although the slope already tends to be midway between controls and subjects with dementia (both AD and DOPA− PD dementia), suggesting that first disruptions are emerging. A closer look hints on a possible subpopulation of those that are closer to controls and those that are closer to subjects with dementia (with more correlated Hcy and Cys), potentially future converters, although this is a mere speculation. Differential expression analysis already shows a trend of reduced betaine (*p* = 0.032, FDR not significant) in PD subjects with MCI, approaching the levels of subjects with dementia, while the values are normal in cognitively unimpaired PD subjects. Furthermore, SAM levels seem to be increased although non-significantly in cognitively normal PD subjects. This could lead to increased CBS activity and clearance of Hcy, thus minimizing cognitive decline.

### 4.4. The Role of Betaine and Hydrogen Sulfide

While Hcy is considered the toxic intermediate in one-carbon metabolism, we found that betaine levels show the best association with MMSE score, i.e., among all subjects as well as among PD subjects across the spectrum of cognitive impairment alone. The association exhibits a sharp decrease in betaine at the beginning of cognitive decline and later plateaus. It is uncertain whether decreased betaine is only an indicator of extensively used BHMT pathway as an alternative route of Hcy metabolism in pathological conditions or whether it is an active player in the etiology of cognitive decline itself. There is increasing evidence that the role of betaine reaches beyond being just a methylation donor [46,47]. Betaine has been also reported to protect the brain from levodopa-induced oxidative stress in rats [48]. Possibly, there is an interplay between reduced betaine synthesis from dietary choline negatively impacting the BHMT re-methylation capacity and driving Hcy elevation on the one hand and elevated Hcy excessively using BHMT pathway and depleting levels of betaine on the other hand, with both converging toward higher Hcy and lower betaine levels, both of which may negatively contribute to cellular homeostasis and cognitive decline.

Hydrogen sulfide is another signaling molecule with a potentially crucial role in cellular signaling and homeostasis. Its physiological levels are now considered neuroprotective, and its reduction in response to inefficient CBS pathway in AD and PD dementia is pro-inflammatory and may exacerbate the pathology [49,50,51].

### 4.5. Limitations

This study has several limitations. We measured only non-protein-bound Hcy rather than total Hcy that typically shows stronger correlations with its negative effects. However, non-protein-bound Hcy better reflects Hcy surges as in the case of levodopa administration. Furthermore, any clinical data including the list of medication and dosage were up-to-date at the time of the last pre-mortem visit but not necessarily at the time of death. This could affect the estimated effect of vitamin supplements included in the regression or underestimate levodopa dosage although we do not expect the results to change significantly. The last visit with an update of MMSE and UPDRS-M scores occurred typically 1–2 years prior to death. Lastly, there is somehow a lower number of non-demented PD subjects when considered as separate subgroups of PD with MCI and cognitively unimpaired PD, compared to other groups. This leads to a relatively lower statistical power for these groups, and effects of a comparable size will achieve lower significance than in other groups.

## 5. Conclusions

We present evidence of levodopa-induced elevation of Hcy in the brain cortex of PD subjects with dementia but not in non-demented PD subjects, independently of levodopa dose and disease progression. This temporary Hcy accumulation in the cortex is also more characteristic of the dementia status in PD than histopathological markers of neurofibrillary tangles, senile plaque, and Lewy body stage. There are signs of reduced CBS activation, possibly due to carbidopa-induced vitamin B6 depletion, together with an impaired flow through MTR, the mechanism of which requires further research. These metabolic restrictions that lead to Hcy accumulation after every administration of levodopa–carbidopa may be directly involved in the etiology of dementia in PD. Controlled clinical trials will be needed to validate the benefits of supplementing B6 and potentially other B vitamins and SAM on cognitive function in PD patients. In AD subjects, we see signs of similarly reduced CBS activation. Indeed, increased intake of B vitamins has already proven effective for AD patients in randomized controlled trials [35]. Additionally, we show that betaine is associated with MMSE across the spectrum of cognitive impairment, and, in another study, betaine supplementation provided benefit in a clinical trial in AD [36].

## Figures and Tables

**Figure 1 nutrients-14-00599-f001:**
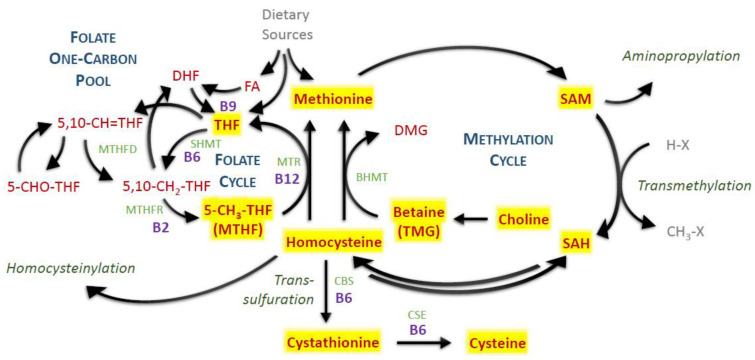
One-carbon metabolism. Red—measured metabolites; yellow highlighted—detected and quantified metabolites; light green—enzymes; purple—vitamin B cofactors. Abbreviations: BHMT, betaine-homocysteine methyltransferase; CBS, cystathionine β-synthase; CSE, cystathionine γ-lyase; DHF, dihydrofolate; DMG, dimethylglycine; FA, folic acid; MTHF, methyltetrahydrofolate; MTHFD, methylenetetrahydrofolate dehydrogenase; MTHFR, methylenetetrahydrofolate reductase; MTR, methionine synthase; SAH, S-adenosylhomocysteine; SAM, S-adenosylmethionine; SHMT, serine hydroxymethyltransferase; THF, tetrahydrofolate; TMG, trimethylglycine; X, methylation acceptor residue.

**Figure 2 nutrients-14-00599-f002:**
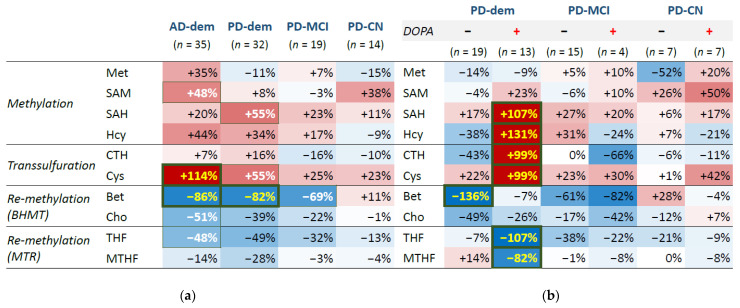
Group heatmap for differential analysis by diagnosis groups. Regardless of current levodopa status (**a**) and with levodopa interaction (**b**) as identified from DOPA concentrations. Red—upregulation; blue—downregulation. Font color: yellow—FDR ≤ 0.05; white—*p*-value ≤ 0.05; black—*p*-value > 0.05.

**Figure 3 nutrients-14-00599-f003:**
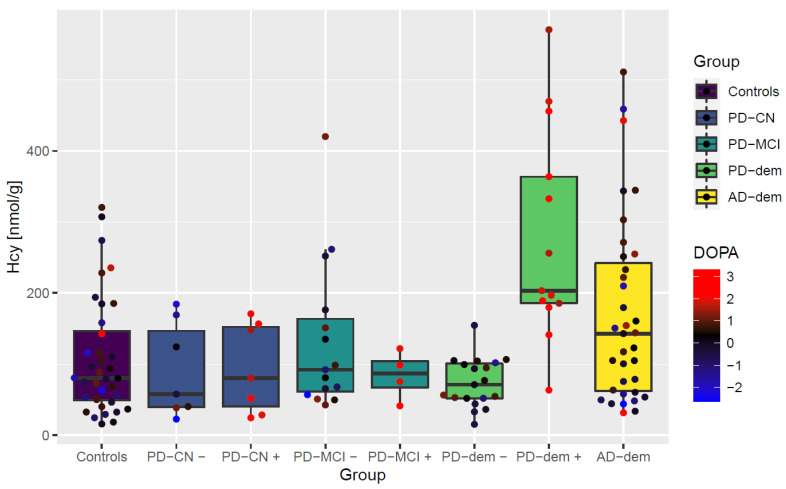
Boxplot of non-protein-bound homocysteine levels by diagnostic group with levodopa interaction. Points represent individual subjects. + and − signs attached as group suffixes denote DOPA+ and DOPA− subgroups. The measured DOPA levels (transformed and standardized) are further visualized as the point color.

**Figure 4 nutrients-14-00599-f004:**
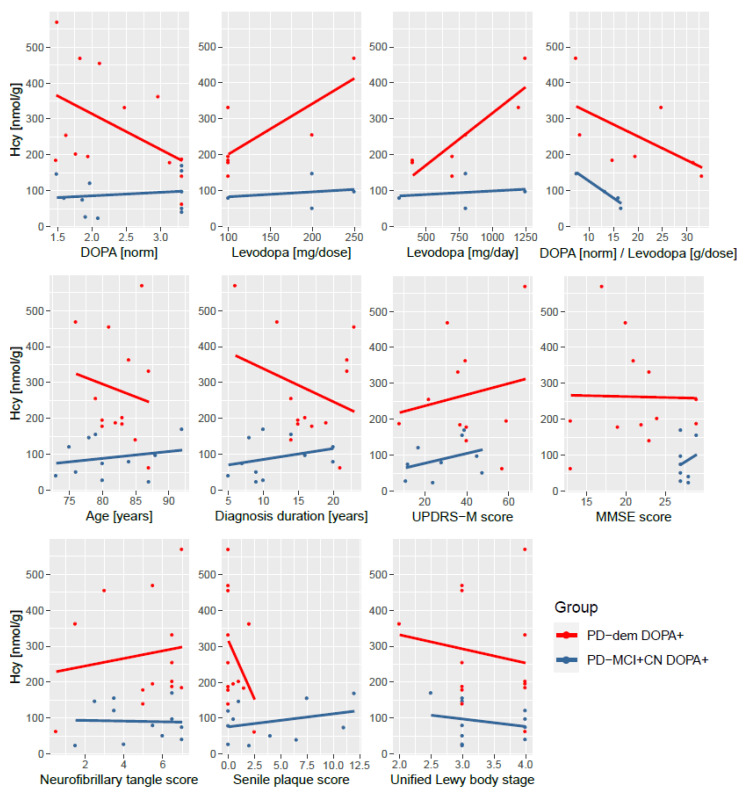
Independence of non-protein-bound homocysteine elevation upon levodopa intake in PD subjects with dementia on medication dosage and measures of disease progression. (**Top**): measured DOPA, levodopa medication amount per dose and per day and a ratio of measured DOPA and levodopa dosage; (**middle row**): age, length of the diagnosis, UPDRS-M motor score, and MMSE cognitive score; (**bottom**): histopathological scores of neurofibrillary tangle density, senile plaque density, and Unified Lewy body stage. The DOPA+ PD subjects with dementia (red) have increased homocysteine levels than non-demented DOPA+ PD subjects (blue) irrespective of these variables. Levodopa dosage, UPDRS-M and MMSE scores are not available for all subjects. Abbreviations: norm, normalized (transformed and standardized).

**Figure 5 nutrients-14-00599-f005:**
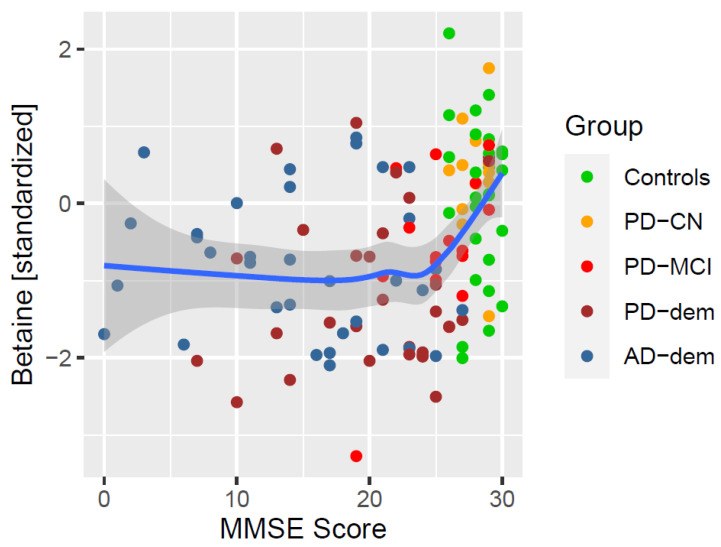
Association of MMSE with betaine. The blue line represents a locally estimated scatterplot smoothing (LOESS) trend with 95% confidence interval shown as the dark gray area.

**Figure 6 nutrients-14-00599-f006:**
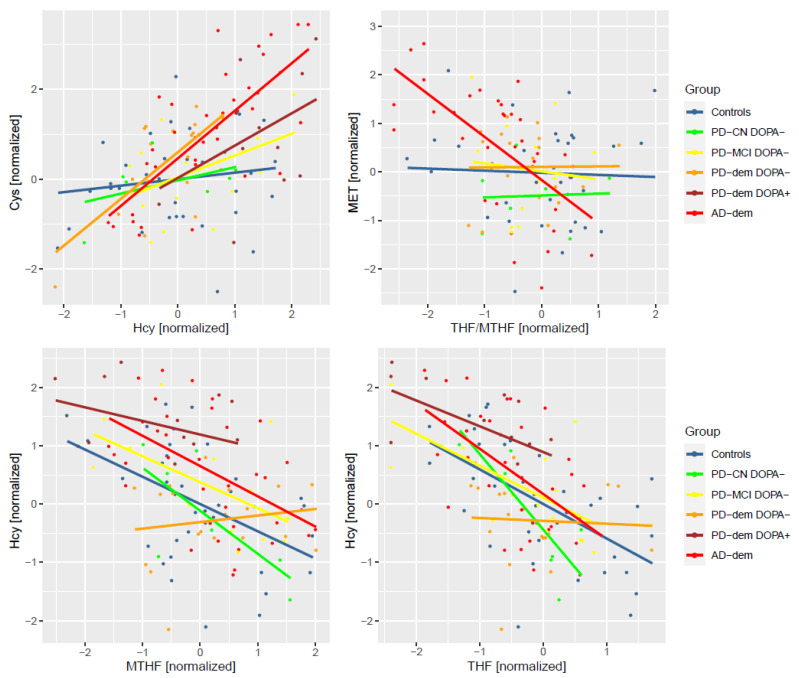
Differentially altered metabolic correlations by diagnosis group. Only DOPA− subjects are included to avoid levodopa-related pathway disturbances. PD-dem DOPA+ subjects are included for comparison with the effects presented for DOPA− PD subjects with dementia to demonstrate their similarity (apart from elevated Hcy) even upon levodopa intake.

**Figure 7 nutrients-14-00599-f007:**
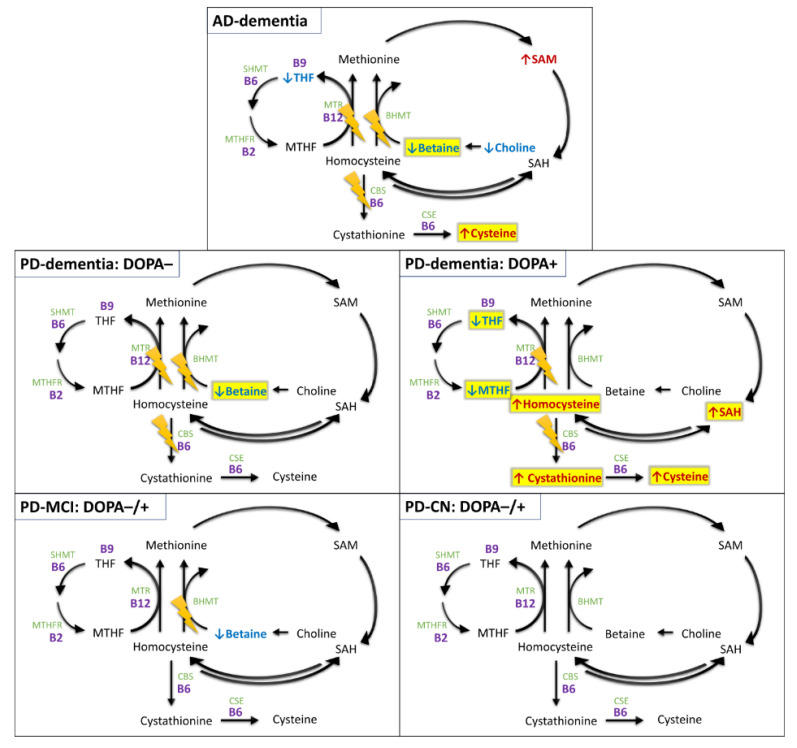
Overview of differentially expressed metabolites and detected pathway abnormalities by group and levodopa status. Upregulated (red) and downregulated (blue) metabolites with *p*-value ≤ 0.05; additionally highlighted with yellow for FDR ≤ 0.05. The lightning icons denote potentially impacted pathways (regardless of primary causality) based on abnormalities found in differential analysis and correlation analysis. Notice that the alterations revealed in DOPA− PD subjects with dementia are well reflected in metabolite concentrations after levodopa-induced Hcy challenge. Light green—enzymes; purple—vitamin B cofactors. Abbreviations: BHMT, betaine-homocysteine methyltransferase; CBS, cystathionine β-synthase; CSE, cystathionine γ-lyase; MTHF, methyltetrahydrofolate; MTHFR, methylenetetrahydrofolate reductase; MTR, methionine synthase; SAH, S-adenosylhomocysteine; SAM, S-adenosylmethionine; SHMT, serine hydroxymethyltransferase; THF, tetrahydrofolate.

**Table 1 nutrients-14-00599-t001:** Subject characteristics.

	Controls (*n* = 36)	AD-Dem (*n* = 35)	PD-Dem (*n* = 32)	PD-MCI (*n* = 19)	PD-CN (*n* = 14)	*p* ^1^
Sex, male, no. (%)	22 (61%)	20 (57%)	24 (75%)	14 (74%)	8 (57%)	0.47
Age, years, mean (SD)	82 (10)	81 (9)	80 (5)	82 (6)	85 (6)	0.1
Race	All White Americans					1
*APOE*, ε4 carrier, no. (%)	5 (14%)	18 (51%)	3 (9%)	3 (16%)	2 (14%)	<0.001
BMI ^2^, kg/m^2^, mean (SD)	25 (5)	24 (4)	27 (9)	25 (8)	22 (4)	0.08
MMSE ^2,3^, points, mean (SD)	28 (1)	15 (8)	20 (6)	25 (3)	28 (1)	<0.001
UPDRS-M ^2,4^, points, mean (SD)	7 (5)	17 (18)	50 (16)	31 (15)	32 (15)	<0.001
Diagnosis duration, years, mean (SD)	N/A	9 (4)	18 (9)	12 (6)	11 (7)	<0.001
Neurofibrillary tangles, density score, mean (SD)	3 (2)	14 (2)	5 (2)	5 (2)	5 (3)	<0.001
Senile plaque, density score, mean (SD)	3 (4)	14 (1)	2 (3)	5 (6)	5 (5)	<0.001
Unified Lewy body stage, score, mean (SD)	0 (0)	0 (0)	3.4 (0.6)	3.7 (0.5)	3.0 (0.5)	<0.001
Vitamin B12 ^2^, supplementing, no. (%)	3 (8%)	2 (6%)	5 (16%)	2 (11%)	2 (14%)	0.67
Multivitamin ^2^, supplementing, no. (%)	17 (47%)	8 (23%)	10 (31%)	6 (32%)	9 (64%)	0.047
Post-mortem interval, hours, mean (SD)	3.2 (1.0)	3.4 (0.8)	3.4 (0.9)	3.4 (1.2)	3.3 (0.9)	0.87
Freezer storage, years, mean (SD)	11 (4)	8 (2)	9 (4)	7 (4)	9 (4)	<0.001

Abbreviations: AD, Alzheimer’s disease; APOE, apolipoprotein E; BMI, body mass index; CN, cognitively normal; dem, dementia; MCI, mild cognitive impairment; MMSE, Mini-Mental State Examination; N/A, not applicable; PD, Parkinson’s disease; SD, standard deviation; UPDRS-M, motor section of the Unified Parkinson’s Disease Rating Scale. ^1^ Group comparison was performed with Fisher’s exact test (count variables) and ANOVA test for unequal group variance (continuous variables). ^2^ Clinical information as recorded at the time of the last pre-mortem visit. ^3^ MMSE values are available for 87% of subjects. ^4^ UPDRS-M values are available for 58% of subjects and were measured for PD subjects when levodopa was not in effect.

**Table 2 nutrients-14-00599-t002:** Measured concentrations by group.

	Controls	AD	PD-Dem	PD-MCI	PD-CN
THF, pmol/g, mean (SD)	144 (78)	104 (50)	106 (63)	118 (64)	128 (36)
MTHF, pmol/g, mean (SD)	199 (106)	217 (108)	188 (110)	217 (110)	201 (93)
SAM, nmol/g, mean (SD)	12.7 (5.0)	15.4 (4.7)	13.1 (4.6)	13.0 (5.6)	14.1 (5.7)
SAH, nmol/g, mean (SD)	13.6 (3.2)	14.4 (3.0)	15.5 (3.0)	14.3 (3.0)	13.5 (2.7)
Met, nmol/g, mean (SD)	80.7 (36.7)	105 (56.8)	80.6 (28.0)	96.3 (69.5)	77.6 (40.2)
CTH, nmol/g, mean (SD)	608 (571)	578 (424)	779 (812)	538 (466)	499 (435)
Bet, nmol/g, mean (SD)	49.1 (31.8)	27.5 (18.3)	27.9 (25.4)	28.4 (18.3)	52.0 (29.4)
Cho, nmol/g, mean (SD)	213 (168)	116 (61)	139 (75)	144 (67)	179 (72)
Hcy, nmol/g, mean (SD)	109 (83)	171 (130)	157 (140)	123 (97)	93 (63)
Cys, nmol/g, mean (SD)	2981 (1214)	4596 (2260)	3787 (1553)	3255 (1163)	3192 (1203)
DOPA, pmol/g, mean (SD)	38 (25)	59 (124)	744 (3337)	238 (603)	659 (1257)

Abbreviations: Bet, betaine; Cho, choline; CTH, cystathionine; Cys, cysteine; DOPA, dihydroxyphenylalanine; Hcy, homocysteine; Met, methionine; MTHF, 5-methyltetrahydrofolate; SAH, S-adenosylhomocysteine; SAM, S-adenosylmethionine; THF, tetrahydrofolate.

**Table 3 nutrients-14-00599-t003:** Differential analysis with a multivariable linear regression model.

	AD-Dem				PD-Dem				PD-MCI				PD-CN			
	β	95% CI	*p* ^1^	FDR	β	95% CI	*p* ^1^	FDR	β	95% CI	*p* ^1^	FDR	β	95% CI	*p* ^1^	FDR
THF	−48%	(−93; −3%)	0.040	0.09	−49%	(−98; 0%)	0.051	0.13	−32%	(−91; 26%)	0.28	0.84	−13%	(−58; 31%)	0.56	0.95
MTHF	−14%	(−65; 38%)	0.61	0.68	−28%	(−80; 25%)	0.30	0.43	−3%	(−62; 56%)	0.92	0.92	−4%	(−60; 52%)	0.88	0.95
SAM	48%	(1; 95%)	0.046	0.09	8%	(−37; 53%)	0.73	0.73	−3%	(−61; 55%)	0.92	0.92	38%	(−29; 105%)	0.27	0.95
SAH	20%	(−28; 69%)	0.42	0.52	55%	(9; 101%)	0.020	0.10	23%	(−34; 80%)	0.43	0.84	11%	(−46; 68%)	0.70	0.95
Met	35%	(−20; 90%)	0.22	0.31	−11%	(−57; 36%)	0.65	0.72	7%	(−52; 66%)	0.82	0.92	−15%	(−88; 59%)	0.70	0.95
CTH	7%	(−40; 54%)	0.77	0.77	16%	(−36; 69%)	0.54	0.68	−16%	(−73; 41%)	0.59	0.84	−10%	(−72; 53%)	0.76	0.95
Bet	−86%	(−137; −35%)	0.001	0.007	−82%	(−136; −28%)	0.004	0.035	−69%	(−132; −7%)	0.032	0.32	11%	(−48; 69%)	0.72	0.95
Cho	−51%	(−96; −6%)	0.028	0.09	−39%	(−85; 7%)	0.10	0.19	−22%	(−68; 24%)	0.36	0.84	−1%	(−51; 48%)	0.95	0.95
Hcy	44%	(−8; 96%)	0.10	0.17	34%	(−15; 83%)	0.17	0.29	17%	(−37; 71%)	0.54	0.84	−9%	(−73; 55%)	0.79	0.95
Cys	114%	(54; 173%)	<0.001	0.003	55%	(1; 109%)	0.050	0.13	25%	(−32; 81%)	0.39	0.84	23%	(−40; 86%)	0.48	0.95

Abbreviations: See previous tables. ^1^ Test for linear regression coefficients with unequal group variance, adjusted for covariates as listed in the Methods.

**Table 4 nutrients-14-00599-t004:** MMSE associations.

	MMSE (All Subjects)				MMSE (PD Subjects)			MMSE (PD Subjects, DOPA Interaction)				
	β	95% CI	*p* ^1^	FDR	β	95% CI	*p* ^1^	FDR	β	95% CI	*p* ^1^	FDR
THF	3%	(1; 5%)	0.010	0.034	2%	(−2; 7%)	0.30	0.50	1%	(−4; 5%)	0.82	0.98
MTHF	2%	(0; 5%)	0.10	0.14	1%	(−4; 6%)	0.67	0.67	0%	(−5; 5%)	0.99	0.99
SAM	−3%	(−5; −1%)	0.014	0.035	−4%	(−8; 1%)	0.10	0.33	−4%	(−9; 0%)	0.09	0.29
SAH	−2%	(−4; 1%)	0.17	0.19	−2%	(−6; 3%)	0.48	0.54	0%	(−4; 4%)	0.88	0.98
Met	−1%	(−3; 2%)	0.64	0.64	2%	(−3; 6%)	0.44	0.54	1%	(−4; 6%)	0.58	0.98
CTH	−2%	(−4; 0%)	0.08	0.13	−4%	(−9; 2%)	0.17	0.43	−1%	(−6; 4%)	0.76	0.98
Bet	4%	(1; 7%)	0.005	0.026	8%	(3; 14%)	0.002	0.021	8%	(3; 13%)	0.002	0.019
Cho	2%	(−1; 4%)	0.16	0.20	5%	(1; 9%)	0.027	0.13	4%	(0; 8%)	0.08	0.29
Hcy	−3%	(−6; −1%)	0.018	0.035	−3%	(−8; 2%)	0.27	0.50	1%	(−4; 5%)	0.80	0.98
Cys	−5%	(−8; −2%)	<0.001	0.009	−2%	(−8; 3%)	0.39	0.54	−2%	(−7; 3%)	0.48	0.98

Abbreviations: See previous tables. ^1^ Test for linear regression coefficients, adjusted for covariates as listed in the Methods (Section 2.4).

## Data Availability

The data presented in this study are available in Appendix A.

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
