# Peer review of "One-Carbon Metabolism in Alzheimer’s Disease and Parkinson’s Disease Brain Tissue"

_nutrients, 2022, doi:10.3390/nu14030599_

Round 1

Reviewer 1 Report

In the present article by Kalecky et al. with the title "One-Carbon Metabolism in Alzheimer's Disease and Parkinson's Disease Brain Tissue", the authors provide a thorough statistical analysis of the concentrations of one-carbon metabolites in human brain samples and their association with various subject characteristics such as a stage of cognitive impairment, levodopa dosage, histopathological density scores, and others. I would like to highlight the authors' closest attention to the limitations of the research and randomization of the samples to avoid any bias. The paper is well organised. Still, several minor issues should be addressed to finalize it.

Subsection 3.2 "Concentrations" is very short (lines 226-229). Since it describes the subjects, it could be united with subsection 3.1 "Subject Characteristics". In addition, the next subsection "Differential Analysis" is also numbered 3.2.

The authors have discovered several interesting associations and correlations, which are described in "Discussion" and "Conclusion". For a better understanding of them, I would recommend adding one more figure where the major associations would be shown. Figure 1 can be used as a template where the pathways disruptions or up-/down-regulation in AD and PD could be indicated.

The abbreviations CI and FDR in the footnote of Table 2 (lines 231-232) are not necessary.

Table 3 possesses a small size of the font. I would recommend reorganizing the table to increase the font if it is possible.

From what I understand, only PD subjects took levodopa so it does not make sense to add AD subjects in Figure 2B and the points indicating DOPA levels in the AD-dem group in Figure 3. If the authors want to report DOPA levels in all investigated subjects, DOPA concentrations could be added in Table 2 (and in Figure 1, if appropriate).

In Supplementary Material A, is PD means PD-CN group of subjects? If it does, I would recommend replacing 'PD' with 'PD-CN'. What unit of measurement was used for APOE in Supplementary Material A?

In Figure S1 the range of 'Frequency' is different for 'Controls' (0 to 6) and 'PD' (0 to 7). I would recommend the authors to make it the same for ease of comparison.

Author Response

Subsection 3.2 "Concentrations" is very short (lines 226-229). Since it describes the subjects, it could be united with subsection 3.1 "Subject Characteristics". In addition, the next subsection "Differential Analysis" is also numbered 3.2.

Response: We have merged the subsection Concentrations into Subject Characteristics as you suggest.

The authors have discovered several interesting associations and correlations, which are described in "Discussion" and "Conclusion". For a better understanding of them, I would recommend adding one more figure where the major associations would be shown. Figure 1 can be used as a template where the pathways disruptions or up-/down-regulation in AD and PD could be indicated.

Response: We have added Figure 7 with summary of our results.

The abbreviations CI and FDR in the footnote of Table 2 (lines 231-232) are not necessary.

Response: We have removed the definition of these abbreviations.

Table 3 possesses a small size of the font. I would recommend reorganizing the table to increase the font if it is possible.

Response: We have simplified the notation of confidence intervals and increased the font size (from 8pt to 9pt). We have made a similar change in Table 4.

From what I understand, only PD subjects took levodopa so it does not make sense to add AD subjects in Figure 2B and the points indicating DOPA levels in the AD-dem group in Figure 3. If the authors want to report DOPA levels in all investigated subjects, DOPA concentrations could be added in Table 2 (and in Figure 1, if appropriate).

Response: We have followed the suggestion and excluded AD group from Figure 3 (originally it was included for transparency, because the regression model still included the AD group and the AD coefficients might have been indirectly impacted due to a potentially different estimation of the other effects should they have any DOPA interaction). We have added DOPA concentrations into Table 2 and Supplementary Material A. During this change, we have noticed a slight inaccuracy in calculation of DOPA concentrations in one of the plates and corrected it. This modification lead to a change in the DOPA+/- assignment for two subjects and we have recomputed all related results accordingly. The overall results and conclusions remain unaffected.

In Supplementary Material A, is PD means PD-CN group of subjects? If it does, I would recommend replacing 'PD' with 'PD-CN'. What unit of measurement was used for APOE in Supplementary Material A?

Response: Yes, this is a good catch. We have corrected the group labels in the Supplementary Material A and properly label the APOE genotype as ε alleles.

In Figure S1 the range of 'Frequency' is different for 'Controls' (0 to 6) and 'PD' (0 to 7). I would recommend the authors to make it the same for ease of comparison.

Response: We have synchronized the axis ranges for both histograms.

Reviewer 2 Report

The present study explored alterations in one-carbon metabolism in Alzheimer’s disease (AD) and Parkinson’s disease (PD) directly in human brain frontal cortex by applying targeted liquid chromatography tandem mass spectrometry (LC-MS/MS) to analyze post-mortem samples obtained from 136 subjects (35 AD, 65 PD, 36 controls). The results showed that elevated cysteine (Cys, +114%) and decreased betaine (-86%) in AD cortex, and decreased betaine (-82%) in PD subjects with dementia (FDR ≤ 0.05), as well as increased S-adenosylmethionine (SAM, +48%), decreased choline (-51%) and tetrahydrofolate (THF, -48%) in AD subjects, increased Cys (+55%) and S-adenosylhomocysteine (SAH, +55%) in PD subjects with dementia, and decreased betaine (-69%) in PD subjects with MCI (p-value ≤ 0.05 but less significant FDR). Further differential analysis with levodopa interaction in PD subjects with dementia showed that upregulation of S-adenosylhomocysteine (SAH, +104%), homocysteine (Hcy, +134%), cystathionine (CTH, +105%) and Cys (+96%) and down-regulation of THF (-117%) and 5-methyltetrahydrofolate (MTHF, -89%) (FDR <0.05). Based on these data, the authors suggested a reduced activity of cystathionine β-synthase (CBS, a vitamin B6-dependent enzyme for the conversion of Hcy into CTH) in AD and PD subjects with dementia, the latter seemingly accompanied by a restricted re-methylation flow via methionine synthase (MTR). The present study also found that only betaine showed significant positive association when metabolic associations with MMSE score in PD subjects were analyzed. There are some concerns as listed in the following:   

(1) The authors pointed out that concentrations of metabolites are often correlated and dependent on each other along metabolic pathways and hypothesized that in case of pathway disruptions such correlations might be considerably weaker, non-existent, or even new ones might appear. As the authors concluded a reduced activity of cystathionine β-synthase (CBS, a vitamin B6-dependent enzyme for the conversion of Hcy into CTH) in AD and PD subjects with dementia, then it is unclear why the present differential correlation analysis still showed a strong correlation between Hcy and Cys in AD subjects and DOPA– PD subjects with dementia. In Figure 2b, the data showed Hcy (+43%), CTH (+8%) and Cys (+114%) in AD-dem group, but Hcy (-25%), CTH (-31%) and Cys (+31%) in PD-dem DOPA- group. It is also unclear why Figure 6 has no Hcy/Cys correlation data for PD-dem DOPA+ group, in which Hcy (+134%), CTH (+105%) and Cys (+96%).

(2) Explain why significantly increased SAH (a potent methyltransferase inhibitor) only in PD-dem DOPA+ group but not in PD-MCI DOPA+ group and PD-CN DOPA+ group (Figure 2b)

(3) Figure 2: The font color for THF in PD-dem group should be white (P=0.05 in Table 3)

Author Response

(1) The authors pointed out that concentrations of metabolites are often correlated and dependent on each other along metabolic pathways and hypothesized that in case of pathway disruptions such correlations might be considerably weaker, non-existent, or even new ones might appear. As the authors concluded a reduced activity of cystathionine β-synthase (CBS, a vitamin B6-dependent enzyme for the conversion of Hcy into CTH) in AD and PD subjects with dementia, then it is unclear why the present differential correlation analysis still showed a strong correlation between Hcy and Cys in AD subjects and DOPA– PD subjects with dementia. In Figure 2b, the data showed Hcy (+43%), CTH (+8%) and Cys (+114%) in AD-dem group, but Hcy (-25%), CTH (-31%) and Cys (+31%) in PD-dem DOPA- group.

Response: We have provided additional explanation for this relationship in the text:

The Cys upregulation and strong significant correlation between Hcy and Cys can result from reduced flow, however counterintuitive it might appear. This is because a restriction and higher saturation will lead to continuous synchronous flow through the pathway instead of immediate Hcy clearance when needed.”

It is also unclear why Figure 6 has no Hcy/Cys correlation data for PD-dem DOPA+ group, in which Hcy (+134%), CTH (+105%) and Cys (+96%).

Response: In Figure 6, we intentionally show only DOPA– subjects by default, so as not to include any levodopa-related disruptions. However, in the bottom part of Figure 6 we do include PD-dem DOPA+ subjects for comparison to confirm the trend is present for all PD-dem subjects. Following your suggestion, we have included PD-dem DOPA+ subjects even for the Cys/Hcy plot to show that the slope remains similar.

(2) Explain why significantly increased SAH (a potent methyltransferase inhibitor) only in PD-dem DOPA+ group but not in PD-MCI DOPA+ group and PD-CN DOPA+ group (Figure 2b)

Response: The elevation of SAH and Hcy (both interconvertible) in PD-dem DOPA+ subjects and not PD-MCI/CN DOPA+ subjects is one of the main findings of this study. This inability of PD-dem subjects to clear levodopa-induced Hcy excess might be the trigger of dementia in PD patients and we discuss two possible reasons, which may be present simultaneously: 1) PD-dem subjects seem to have an impeded re-methylation pathway via MTR along with overutilized/depleted re-methylation pathway via BHMT. We do not see these changes in PD-MCI/CN subjects – only a starting disruption in BHMT pathway in PD-MCI. 2) Less efficient CBS pathway in PD-dem subjects, which should otherwise clear the Hcy excess.

We have added an explanation that SAH and Hcy are interconvertible and we now explicitly mention the possible reasons why PD-MCI/CN subjects might normally clear the Hcy excess contrary to PD-dem subjects (section 4.3).

(3) Figure 2: The font color for THF in PD-dem group should be white (P=0.05 in Table 3)

Response: The value is actually slightly above 0.05 and was shown as 0.05 due to rounding. We have now clarified this by including one more digit – 0.051.